

# The ice-vapor interface during growth and sublimation

Maria Cascajo-Castresana[1,*], Sylvie Morin[1,2], Alexander M. Bittner[1,3]

[1] CIC nanoGUNE (BRTA), Av. Tolosa 76, 20018 Donostia-San Sebastián, ES-20018, Spain
[2] Department of Chemistry, York University, Toronto, ON M3J 1P3, Canada
[3] Ikerbasque Basque Foundation for Science, 48009 Bilbao, ES-48009, Spain

[*] Present Address: Tecnalia Research & Innovation, Paseo Mikeletegi 2, 20009 Donostia-San Sebastián, Spain

*Correspondence to*: Alexander M. Bittner (a.bittner@nanogune.eu)

**Abstract.** We employed Environmental Scanning Electron Microscopy (ESEM) in low humidity atmosphere to study the complete scenario of ice growth, coalescence of crystallites, polycrystalline film morphology and sublimation, in the temperature range of -10 ℃ to -20 ℃. First, individual ice crystals grow in the shape of micron-sized hexagonal columns with stable basal faces. Their coalescence during further growth forms thick polycrystalline films, consisting of large grains separated by grain boundaries. The latter are composed of 1 to 3 µm wide pores, which are attributed to the coalescence of 15  defective crystallite surfaces. Sublimation of isolated crystals and of films is defect-driven, and grain boundaries play a decisive role. A scallop-like concave structure forms, limited by sharp ridges, which are terminated by nanoscale asperities.

## 1 Introduction

Ice covers a much smaller area of our planet than liquid water. Some of this ice is subjected to large seasonal variations in temperature, giving rise to melting and sublimation. Understanding ice melting, flowing, sublimating and evaporating is key 20  in understanding fully the impact of global warming. Here we focus exclusively on the study of ice/vapor interfaces, as present on snow, glaciers, permafrost soil, sea ice, and in clouds. Our low humidity conditions are of special relevance for sublimation at the poles and in Greenland, where ice sheets are in contact with air of low humidity, at temperatures well below those of most glaciers on other continents (Bliss et al., 2011). The sublimation (and the redeposition) rate ultimately determines whether ice fields can exist at all.

The literature on ice growth and nucleation is vast, and well documented in recent reviews (Bailey and Hallett, 2004; Murphy and Koop, 2005; Libbrecht, 2005; Bartels-Rausch et al., 2012; Bartels-Rausch et al., 2014). It covers a broad spectrum of



disciplines related to the study of geology, atmospheric science, and planetary science, but it is usually restricted to the macro- and millimeter scale. We modelled ice/vapor interfaces inside an Environmental Scanning Electron Microscope (ESEM), where temperature and humidity values can simulate conditions found in cold climates. Vapor freezing and ice sublimation

give rise to micro- and nanoscale surface features that we characterize in detail. Freezing and sublimation depend on the microscopic ice morphology, and on its dynamical changes. We thus contribute microscopic evidence for relevant processes in atmospheric physics.

Under some conditions, especially in clouds, small water droplets transform into small nearly spherical ice crystals with well rounded corners and edges. They are known as "droxtals", a term that combines the words droplet and crystal (Takahashi and

Mori 2006, Gonda and Yamazaki 1978). The dense packing of such individual ice crystals results in complex scenarios. In glaciers, the ice crystals are deformed and rotated, ice sublimes and water vapor is redeposited, hence ice crystals constantly form new grain boundaries. The local curvature, but more specifically the exact shape of the grains should profoundly influence sublimation kinetics. An in-situ method such as the one proposed in our study would allow to observe changes in morphology, determine the role of grain boundaries as well as impurities in ice in real time (Gonda and Yamazaki 1978, Vetráková 2019).

Here, we consider solely the solid/vapor interface, for temperatures well below the melting point. However, ice growth from vapor is not restricted to direct incorporation of molecules into the solid phase, it can also proceed via a supercooled liquid phase (Sei and Gonda, 1989), which can be extended (e.g. as droplet), or located at the ice surface as a "quasi-liquid layer". In addition, the ESEM environment could offer flexibility to approach realistic scenarios, such as brine-filled veins in ice (Vetráková et al., 2019).

Ice growth requires that the water vapor reaches conditions of temperature $T$ and water vapor pressure $P_{water}$ for which water freezing and sublimation are at equilibrium. However, ice nuclei formation needs to overcome the activation nucleation barrier, through water vapor supersaturation. Once ice is formed, the solid/vapor interface of ice is subject to freezing (growth, deposition or desublimation) and sublimation (loss of water molecules), depending on temperature and pressure of the vapor phase. Summing up, the ice surface can grow when $P_{water} > P_{sat}$, and it will invariably lose material by sublimation when

$P_{water} < P_{sat}$, where $P_{sat}$ corresponds to the vapor pressure on the sublimation line of the phase diagram. We define the relative humidity $h$ by

$$h = \frac{P_{\text{water}}}{P_{\text{sat}}} \qquad \qquad (1)$$

Thus, sublimation requires humidity values below 100%, while above 100%, i.e., at supersaturation, $P_{water} > P_{sat}$ results in ice growth via deposition (Kiselev et al., 2017). For ambient pressure (and hence slow diffusion in the gas phase, Lamb and Scott

(Lamb and Scott, 2010) provide an overview of experiments and theories for ice growth from 30 °C up to 0 °C. However, by





restricting the water vapor pressure below that of the triple point (611.7 Pa), which is 165 times lower than ambient pressure, conditions of very fast diffusion in the gas phase are achieved (compare also Sei and Gonda, 1989).

The only ice polymorph that plays a role under our low-pressure conditions is the hexagonal ice ($I_h$) structure (at much lower
temperature, cubic ice could be observed). The hexagonal unit cell can arrange in macroscopic hexagonal plates and columns, where the top and bottom faces are the (0001) basal planes, which are perpendicular to the c-axis. Faces parallel to the c-axis are called prism or prismatic faces and are the (1010) and (1210) faces. Another set of planes in $I_h$ are known as pyramidal faces. They are bevelled facets, associated with each prism surface (Pfalzgraff et al., 2010). If water (in air) condenses preferentially on basal faces, an ice crystal adopts a prism shape (hexagonal columns). On the other hand, plates are formed if
prismatic faces are preferred areas for water condensation.

The optimal growth temperature is 6 ℃ for the basal face, and of 13 ℃ for the prism face (at ambient pressure). Sei and Gonda (Sei and Gonda, 1989) provide details for growth in 40 Pa of air, at several % of supersaturation, altogether these conditions are comparable with that achievable in our ESEM. Their results nicely explain some of our data, especially the growth of high
(long) hexagonal columns. Libbrecht discusses similar results (Libbrecht, 2004), quoting mainly systematic effects as problems (latent heat effects, diffusion effects, substrate interactions). In conclusion, some generally accepted rules are: The growth velocity on the prism face is higher at temperature $T > 5$ ℃ (this should result in plates) and at $20$ ℃ $< T < 10$ ℃, on the basal face the rate is faster for $10$ ℃ $< T < 5$ ℃ (this should give columns). This is in accordance with Bailey and Hallett (Bailey and Hallett, 2009), who suggest the dominance of plates to 4 ℃, columns to 8 ℃, and again plates to 22 ℃. However,
occasionally more complex shapes can be found, similar to "bullet rosettes", which normally appear at temperatures below 40 ℃ (Hallett and Mason, 1958). Our experimental conditions explore the range from -20°C to 0°C, at low water vapour pressure (i.e., below the triple point pressure of 611.7 Pa).

Crystals of ice, especially in clouds, take on a plethora of shapes: Dendritic snowflakes, plates, pillars, needles and columns
(Libbrecht, 2005; Bailey and Hallett, 2009; Takahashi and Mori, 2006; Gonda and Yamazaki, 1978). Such shapes develop by condensation from the gas phase, or, in clouds more frequently, by freezing in supercooled water. They are characterized by sharp edges and facets. They can transform into small nearly spherical ice crystals with well rounded corners and edges, which are known as "droxtals", a term that combines the words droplet and crystal (Gonda and Yamazaki, 1978; Takahashi and Mori, 2006). During further growth (in clouds, or more frequently on snow and ice), the dense packing and finally the coalescence
of individual crystals results in complex scenarios, which ultimately produce polycrystalline material. We here show details of this process on the microscale, separated between single and polycrystalline surfaces.

Why is the microstructure and its change during growth and coalescence (Nair et al., 2018) important? Grain boundaries show up, and in absence of impurities, typical microscale morphologies develop (Libbrecht, 2005; Sazaki et al., 2010; Asakawa et





al., 2014; Kiselev et al., 2017), and rearrange (Krzyzak et al., 2007). Even supercooled water could exist (in highly curved
nanoscale structures) (Nowak et al., 2008). In nature, impurities can concentrate in such defects (Baker et al., 2003; Bartels-
Rausch et al., 2012; Bartels-Rausch, 2014), for example forming networks of water-filled microscale "veins", sometimes of
relevance as habitat for organisms (Mader, 1992; Mader et al., 2006; Buford Price, 2000). Relevant structural details can
already be observed on the nanoscale (Hondoh, 2009). This impurity effect has also been studied (or assumed to play a role)
for micro- and nanoscale imaging (Cross, 1969; Cross, 1971; Mulvaney et al., 1988; Cullen and Baker, 2001; Rosenthal et al.,
2007). Also, in this case, ions (from salts or acids) or dust particles in the crystals are thought to be the underlying cause for
structures that develop during sublimation.

The nature of such morphologies, how they are attained from microcrystals, and how ESEM helps to elucidate them, is topic
of this work, where we studied the deposition and sublimation of isolated ice crystals and of polycrystalline ice films on oxide
surface (oxidized silicon and oxidized copper). The surface morphology on the microscale was then followed in real time using
ESEM (Kiselev et al., 2017; Ebert et al., 2002), during further growth, and during sublimation. This allowed us to access a
variety of growth morphologies, such as columnar isolated crystals, droxtals, and polycrystalline films, as well as to follow
their development during growth and sublimation with high spatial resolution.

## 2. Experimental Section

### 2.1 Environmental Scanning Electron Microscope (ESEM)

Images and movies of ice crystals during growth and sublimation on the wafer pieces were recorded in a FEG (Field Emission
Gun) ESEM (Quanta 250, FEI), operated in the wet mode (Toth et al., 2003). The sample stage temperature was adjusted
between 10 °C and 20 °C by Peltier cooling. However, temperatures down to about -40°C could be achievable, at least for
small samples. Resistive heating allowed to reach ambient temperature quickly after the experimental runs. A thermocouple
was located inside the Peltier stage, hence all temperature values are reported for readings at that location. Potential temperature
gradients from Peltier stage to the substrate (see below) are very small; we verified that sublimation occurs at the correct
temperature, when the pressure is lowered slowly (several Pa/min), and that first surface changes due to ice growth are observed
at the same values when the pressure is increased even slower (1 Pa/min). Large pressure jumps (above ca. 50 Pa) often result
in a short loss of temperature control, due to sublimation or deposition of large amounts of ice.

The imaging gas was pure water vapor, introduced via an automated leak valve. The vapor was produced from a heated
reservoir of water (18 MOhm cm, <5ppb total organic content, Millipore), containing also a Pt wire for the catalytic
decomposition of organic contaminants. The relative humidity in the chamber was controlled by increasing or decreasing the
pressure in small steps (< 5 Pa), in the range from ≈ 50 Pa up to ≈ 700 Pa. Our beam voltage was set to values around 10 kV,
the spot size was 3 to 5 (spot diameters ≈ 1 nm to 7 nm), and the aperture inside the column was kept at its smallest value,
which reduces beam current and thus beam damage. The Supplement provides details on how to avoid beam damage, which



can be mass loss (sublimation) or carbon deposition (from gas contamination) and melting (heating). In this way we avoided localized artifacts.

Electron detection was achieved with the large field detector (LFD), placed some cm sideways (in the SEM images this is
always the right-hand side), and away from the pole piece, or with the gaseous secondary electron detector GSED, a thin gold line, which encircled the pressure-limiting aperture. Standard settings were $1024 \times 884$ pixels with dwell times mainly of 10 to 30 µs for real-time images, and about ten times lower for the fastest scans. For a beam current of 1.6 nA, applied to the smallest possible spot of 1 nm diameter, we obtain a maximal electron dose of 14000 e/s (Alonso et al., 2013).

The saturation water vapor pressure over ice, $P_{sat}$, was determined from the temperature reading in the ESEM, according to Murphy and Koop (Murphy and Koop, 2005):

$$P_{\mathrm{sat}}(T)/\mathrm{Pa} = \exp(9.550426 - 5723.265\mathrm{K}/T + 3.53068\ln(T/\mathrm{K}) - 0.00728332T/\mathrm{K}) \tag{2}$$

for $T > 110$ K. The humidity $h$ is then simply the ratio of the measured pressure (which corresponds to $P_{water}$) and $P_{sat}$, see eq. (1). The technical details of our ESEM setup do not allow for stable $h > 1$ conditions. Any pressure (or temperature) adjustment to values in the $P$-$T$ region of ice cause lowering of the pressure (pumping), or increase of temperature (heating of the sample), thus attaining the phase coexistence line at $h = 1$. This automatic pressure control balances water vapor supply and removal by pumping, and allows to reach $h > 1$ only for short times (at best some minutes). Due to the size of the specimen chamber,
most of the vapor is at ≈20 ºC, but due to the very low flow rate, it attains the substrate temperature when it approaches the Peltier stage/sample assembly.

## 2.2 Substrate

The experiments were carried out with an $SiO_2$ film grown on an n-doped Si wafer. The silicon wafer ((111) orientation, thickness 0.27 mm, doped with phosphorus, resistivity 1 Ω cm) is cut in $4 \times 4$ mm² pieces by a wafer dicing saw (Disco
DAD321). The pieces are thoroughly cleaned by sonication in an ultrasonicator (VWR Ultrasonic Cleaner) for 15 min in a sequence of three solvents: isopropanol (LC-MS chromasolv®, Sigma-Aldrich), acetone (ACS reagent >99.5%, Sigma-Aldrich), water (18 M Ω cm, < 5ppb total organic content, Millipore). They are blown dry with nitrogen and surface-oxidized in an oxygen plasma cleaner (Femto Diener) for approximately 8 min with a 10 sccm gas flow (10 Pa oxygen pressure). This surface treatment renders the wafers hydrophilic. For an experiment, liquid gallium-indium alloy is employed for bonding the
wafer to a home-made copper stub. The stub is tightly fit into the Peltier stage by an aluminium foil spacer.





## 3. Results and discussion

While clean flat oxidized Si wafers (see Experimental Section) are typically used in our work, relatively rough copper surfaces (exposed to ambient air and thus oxidized) were also suitable substrate to grow ice. Both substrate oxidized surfaces are amorphous. In our experiments, the pressure is adjusted to < 200 Pa ($h \ll 1$), then the substrate temperature is lowered from ambient to the desired ice growth temperature ($\geq$ -20ºC). The pressure was then slowly increased until $h = 1$. The rate of ice growth can be controlled by controlling the humidity in the chamber, for example, fast growth can be achieved by pressure jumps to $h > 1$ (see Supplement), and stabilization at $h \leq 1$. The idea is to keep isobaric and isothermal conditions. However, small changes in sample temperature, which we carefully monitored, cannot be avoided. Subsequent sublimation requires $h <$ 1, again accompanied by slight temperature changes.

In air, we would expect plate morphology between 20 ºC and 10 ºC under our low-pressure conditions (i.e. at fast gas phase diffusion), but only for $h \gg 1$, at much slower gas phase diffusion, dendritic snowflake crystals (Gonda and Yamazaki, 1978; Libbrecht, 2005; Takahashi and Mori, 2006; Bailey and Hallett, 2009). Between 10 ºC and -5°C, crystals can grow as columns or needles. We found for low supersaturation mainly hexagonal plates and columns from 20 °C to 10 °C (as discussed below). In accordance with Bailey and Hallett (Bailey and Hallett, 2004), individual crystals are of hexagonal geometry with facets of equal lengths as well as hexagonal scalene crystals, with three dissimilar facet lengths.

### 3.1 Early stages of growth

By increasing the water vapor pressure (by 10% - 20%) rapidly (in less than 5 s, see Supplement), the growth of individual hexagonal ice crystal is achieved on the oxidized Si wafer, i.e. on an amorphous, OH-terminated SiO$_2$ surface. On the substrate, the orientation of ice crystals is random, with a few crystals having their c-axis perpendicular to the surface plane. The ice nucleation is progressive and homogeneous for the pressure and temperature conditions reported here. A characteristic example of ice morphology is given in **Figure 1A**, where randomly oriented isolated ice crystals are clearly visible, with a few features displaying highly defective crystal aggregates with well-resolved grain boundaries. These aggregates originate from the merging of nearby crystallites. The bright ridges in the image are usual for SEM (Castle and Zhdan, 1997; Nair et al., 2018) and correspond to a locally high flux of secondary electrons, due to higher emission rates from surfaces that point towards the detector. This is also the case for the prismatic faces on the left of **Figure 1B**. This image also shows another typical effect, namely that flat surfaces appear to have a bright, but blurry rim, in the vicinity of sharp edges. This is due to negative charges, which accumulate preferentially at sharp edges and corners. ESEM reduces these charges but cannot eliminate them completely.

However, only a minority of the isolated crystals are oriented vertically, with the basal plane parallel to the surface (Nair et al., 2018). This contrasts with ice growing on crystalline AgI (Bryant et al., 1960), and with ice nucleating on active sites on





the surface of a type of feldspar (Kiselev et al., 2017), where epitaxial growth is observed. For AgI, the hexagonal crystal
c-axis is oriented perpendicular to the basal plane of AgI, i.e., the basal plane of ice and AgI are parallel. The observation of a
hexagonal morphologies (on our substrate, as well as on AgI and on feldspar) is consistent with the $I_h$ phase of ice. Any other
ice phase would require much higher pressures and/or much lower temperatures. There is only one phase that could be
metastable in our pressure range, namely cubic ice. Thürmer and Nie grew this phase epitaxially on a Pt(111) surface in
ultrahigh vacuum (Thürmer and Nie, 2013), but at much lower temperature (-133 °C), in thin layers of plate-like morphology.

Our conditions result in slow growth rates where water molecules attach more readily to facets with high Miller indices, and
ice growth is homogeneous on the prismatic faces, keeping the edges of crystals straight (Elbaum and Wettlaufer, 1993).
**Figure 1B** shows a well-resolved hexagonal ice crystal with its basal plane (0001) oriented nearly parallel to the substrate.
When the growth is monitored over time a growth rate of $\approx$ 100 nm/s is observed at -18.7°C and 127 Pa, hence at $h = 1.09$ (9%
supersaturation) (for more details see Supplement). Our conditions are best compared with those employed in ESEM studies
by Pedersen et al. (Pedersen et al., 2011) (300 nm/s at -20°C, $h$ not given) and Kiselev et al. (Kiselev et al., 2017) (3000 nm/s
at -22°C and $h = 1.18$). It is worth noting that growth rate data, as well as those from other studies (if available), vary
substantially over time, usually the rate is observed to increase (Pedersen et al., 2011).

Comparison of growth rates with other studies is very difficult, for example, other ESEM studies were carried out at
temperature at or below 30 °C (Pfalzgraff et al., 2010; Magee et al., 2014) and were concerned with qualitatively different
growth modes. Specifically, the influence of air up to ambient pressure reduces the mean free path in the gas phase, and thus
alters the velocity distribution of the water molecules impacting the ice surface. In these studies, even lower growth rates were
reported, which can yield different growth morphology. Above 10°C, the growth rate is much higher, e.g., >10000 nm/s at 7°C
and 450 Pa (Chen et al., 2017). This is based on enhanced gas phase diffusion at the higher temperature, while the mean free
path remains nearly unchanged (Pruppacher and Klett, 1978). Although not related, our growth rates are similar to those
reported at the air/water vapor-ice interface at much higher pressures (Libbrecht and Rickerby, 2013), where the very high
growth rate at relatively high temperature is counterbalanced by diffusion in a much denser gas phase, i.e., resulting in a smaller
mean free path (Pruppacher and Klett, 1978) (see Supplement). A better comparison is possible with ice growth in 40 Pa of
(wet) air, reported by Sei and Gonda (Sei and Gonda, 1989), who find (macroscopic) results in very good agreement with the
growth rate for the feature shown in **Figure 1B**.

Some of the crystals in **Figure 1A** also display rounding-off ("roughening") of their edges (Nelson, 1998), but because not all
crystals start growing at the same time, the effect is not observed consistently. Here and in the remaining of the text we use
"roughening" to describe the development of macroscopically smooth, rounded shape. This process is based on the gradual
disappearance of flat crystal planes (facets) (Nelson, 1998), often accompanied by a change of growth kinetics for crystals
near equilibrium (Elbaum, 1991). It implies a high mobility of surface atoms/molecules (Maruyama, 2005; Krzyzak, 2007;



Asakawa et al., 2014; Magee et al., 2014). The nearly spherical shape of the roughened crystals with smooth corners and edges, is analogous to that of droxtals ("droplet crystals") (Gonda and Yamazaki, 1978; Takahashi and Mori, 2006). These effects are
220 clearly visible in our images where ice crystals have an intermediate morphology between a crystal habit with sharp edges and a completely rounded sphere (see **Figure 1A** and **Figure 2**).

We show examples of these morphologies in **Figure 2**, where, **e**xcept for the hexagonal pillar 5 (see label in Figure 2), all crystals are analogous to droxtals. Crystals 1 and 2 have changed their shape significantly during growth. In the droxtal-like crystals 1 and 2 the basal planes are neither perpendicular nor parallel to the surface and correspond to a later stage of growth.
For crystals 3 and 4 less time has elapsed since nucleation and they still exhibit prism-like morphology, with basal planes oriented perpendicular to the surface in agreement with Gonda and Yamazaki (Gonda and Yamazaki, 1978). Their morphology is analogous to droxtals grown in water, as the rounding-off (roughening) occurred when the c-axis had approximately the same length as the a- and b-axes (Gonda and Yamazaki, 1978). Crystal 5 formed after the others (crystals 1-4) and does not display any roughening at the time the image was recorded.

Pfalzgraff et al. (Pfalzgraff et al., 2010) found pyramidal facets and curved (termed by the authors as "conical non-facetted") surfaces associated with ice growth. Pyramidal facets were not found, consistent with our observations, but rather smooth and curved (non-facetted) "roughened" (Elbaum, 1991) surfaces (as shown in **Figure 2**)). Pfalzgraff et al. grew their ice crystals typically between -30 and -45°C, and report growth rates as fast as 700 nm/s for the prismatic faces. They also used the onset of ice ablation as a function of temperature during a variable pressure SEM experiment to provide a way to relate the conditions
in the SEM chamber to establish the frost point. Because pressure changes are reported as ratios of water vapor pressure at the ablation temperature over water vapor pressure at the cooling stage temperature, further comparison with our experimental conditions is difficult, in contrast to ESEM studies at higher temperatures (Nair et al., 2018; Chen, 2017).

We measured local growth velocities at isolated and merging crystals shown in **Figure 3** (see Supplement). For our set of parameters (moderate supersaturation) we obtain values up to 200 nm/s. The values can be separated into max 100 nm/s at the
240 six edges of the basal plane, and 50 to 150 nm/s for some prism faces. Some edges, though, appear to be completely pinned, which is not due to contact with the substrate. However, some faces (e.g., the basal plane A1 in **Figure 3**) expand only very slowly at their edges, and practically not in their normal. This is ultimately responsible for the droxtal-like shape, where the basal plane is almost preserved, while all edges become progressively rounded until they merge. Even then, the center of the basal plane are preserved, as seen in **Figure 4A**.

When two growing crystals merge on the substrate, their relative orientation is usually at an angle, and a grain boundary forms. In the top view, a grain boundary appears as a long dark stripe of some μm thickness (see **Figure 3A** (before merging) and **Figure 3B** (after merging). In addition, upon contact with other crystals, an isolated crystal can develop a large amount of grain boundaries on some facets during growth (see **Figure 3B and 3C**). These grain boundaries are formed very fast and



appear even hundreds of µm away from the area of contact between two crystals. The interface between two planes with different orientations then becomes the nucleation site for additional molecular layers that are not matching with the structure found initially on A1 and B1 (See **Figure 3B** for example). When several new molecular layers nucleate at different positions on a crystal, its morphology is altered significantly and several grain boundaries and dislocations are identifiable (for additional example see also **Figure 1A**). Merging of ice crystals and formation of grain boundaries was also observed with ESEM, reported by Magee et al. (Magee et al., 2014) at much lower temperatures and pressures (below -20°) and by Pedersen et al. (Pedersen et al., 2011) at very similar to our conditions of pressure and temperature. Pedersen et al. propose that an avalanche of dislocations can originate at the newly forming grain boundary (Pedersen et al., 2011), in agreement with our results.

In the early stages of the defective film growth, it was possible to record a process analogous to the onset of spiral growth (macro-steps, see white dashed line in **Figure 3D**). Such growth spirals must extend a minimum distance of tens of nm from a surface in order to be visible in ESEM. Very small spirals can more conveniently be detected at extremely low temperature (-133 °C) (Thürmer and Nie, 2013), where an orientational mismatch in adjacent ice crystals causes defect formation. Translated to the microscale, this could explain the high density of domain boundaries and screw dislocations observed in our much thicker ice films. Larger growth spirals develop under atmospheric conditions of growth (in water vapor/nitrogen at -10 °C to -15 °C), as found by optical microscopy techniques (Sazaki et al., 2010; Sazaki et al., 2014; Asakawa et al., 2014).

Hence, our results are in agreement with well-known results at low pressures and at low supersaturation (Kiselev et al., 2017; Sei and Gonda, 1989). Higher pressures, i.e., admixture of air, could be an option for a high-pressure cell; higher supersaturations are a principal issue with these conditions are the fast growth rates and size of resulting features that are more suitable for optical microscopy.

**3.2 Growth of polycrystalline ice films**

Crystal aggregation yields morphologies that differ much from the general morphology of isolated crystals, at least in the early stages of growth (see **Figure 1**). Merging of many isolated crystals (see **Figure 3** for an example of three crystallites) produces a polycrystalline surface (see **Figure 4A**, where some crystals have merged already). Ultimately, even the basal planes merge with other crystals, many grain boundaries appear, and facets are no longer visible (**Figure 4B**). The formation of a continuous film of ice required at least tens of µm thick deposits. Such polycrystalline ice films show typical dark bands, which are assigned to grain boundaries. The grain boundaries display typical Y shapes indicative of a nearby dislocation (see **Figure 4B**). This is in agreement with Krausko et al. (Krausko et al., 2014) who froze liquid water inside the ESEM specimen chamber, but also with ESEM of condensing water vapor (Nair et al., 2018; Chen et al., 2017). Grains can be nicely detected and assigned by EBDS (Montagnat et al., 2015), but measuring the depth of the grain boundaries requires AFM. The typical depth range was found to be 0.05 to 0.5 µm (Krzyzak et al., 2007; Zepeda et al., 2001). We found larger values of up to 10 µm for the



depth of grain boundaries (see Supplement), which could be due to differing conditions between studies, such as observation at a later stage of growth. The width of the grain boundaries was 2 µm on average.

The grain boundaries appear, at first glance, similar to veins in natural ice (Mader, 1992; Mader et al., 2006; Buford Price, 2000), and indeed our structures sometimes meet at angles around 120º (**Figure 4**). However, the veins are much more frequently oriented at relative angles of 120º, they are larger, and they are filled by brine (salt water of low freezing point).

### 3.3 Microscale "pores" inside the grain boundaries

During our measurements on polycrystalline ice films, we observed features that appear as small circular dark spots, which we term "pores", aligned inside the grain boundaries (which are not to be confused with "ice veins" (Mader, 1992,; Mader et al., 2006; Buford Price, 2000). Analogous features to our pores have been observed at ice/brine interfaces (Vetráková et al., 2019). The pores appear in ESEM as black dots (see **Figure 4**) of 1 to 3 µm in diameter, which corresponds to the width of the grain boundaries. Under the chosen imaging conditions, pores appear in all the grain boundaries in **Figure 4**, but they are not evenly 295 distributed. These features are not due to beam damage (see our detailed discussion in the Supplement). Furthermore, under our experimental conditions it is unlikely that the pores are related to the veins reported in salt-contaminated ice (Mader, 1992,; Mader et al., 2006; Buford Price, 2000) (see introduction). It is the first time these features are reported for similar growth conditions.

We postulate that pores form when the facet of at least one crystal with an uneven surface (such as A2 in **Figure 3**) merges with another facet, see (see Supplement), a complex scenario that forms the base for polycrystalline growth from coalescing crystallites. The uneven surface would then result in a large number of pores. Their diameter along a given grain boundary would be related to the height of the uneven features. The situation is more complex when two uneven surfaces meet: Now curved grain boundaries form, filled by pores (**Figure 4**) (model see Supplement). Occasionally observed larger holes (>10 305 µm size, see **Figure 4** for two characteristic examples) can originate from large defects produced when growing grains merge laterally, while their vertical growth rate is too fast to allow for large gaps between the grains to be filled. Similar, but much smaller holes are known from epitaxial ice growth (Thürmer and Nie, 2013). The nanoscale morphology of the grain boundaries is unknown, and beam damage precludes further analysis by ESEM (see Supplement).

The circular shape of the pores might be surprising: given the low temperatures we applied during growth, coarsening towards a smooth shape should not readily occur. However, the presence of high local curvature and (possibly) irregular sub-microstructures would translate into high rates of sublimation and re-adsorption, but only on a local scale. Diffusion of water molecules in the gas phase and on the solid surface grain boundaries would lead to local roughening and coarsening resulting in rounded, nearly circular shape, without substantial evaporation.



## 3.4 Sublimation of isolated ice crystals

Sublimation is achieved by lowering the pressure to h < 1. ESEM imaging turned out to be very good when lowering the pressure by at least a few Pa. In the absence of defects such as dislocation and grain boundaries, water molecules can detach more readily from crystal edges where the prismatic and basal faces meet, as well as from the prismatic faces. **Figure 5A** shows this effect, and the resulting apparent slight roughening of the edges. Clear changes in morphology also take place on top of the basal face where an uneven morphology develops. For our temperature range (20 ºC to 10 ºC), crystals of sizes in the 100 μm scale evaporate in several min, as found by Nelson with optical microscopy (Nelson, 1998). More detailed, SEM and ESEM, e.g. in **Figure 5**, after 7 min the hexagonal plate-like crystal shrank both in height and diameter, and it completely disappeared in ≈ 13 min. The lateral sublimation rate was initially slow, 80 nm/s between **Figure 5B** and **Figure 5C**, but then increased rapidly, as the crystal surface attained an irregular morphology, and as its thickness decreased. The estimated rate between **Figure 5C** and **Figure 5D** is 200 nm/s. This value depends on temperature and pressure; related observations on single crystals are documented at 45ºC (Pfalzgraff et al., 2010), but also in our parameter range (Cross, 1969; Magee et al., 2014), however, without quantification. The morphological similarity to our observations is excellent; the typical ridge or "scallop" morphology (Nair et al, 2018; Cross, 1969; Magee et al., 2014) of the prism faces was always observed (**Figures 5C** and **Figure 5F**). An unusual observation is reported in **Figure 5B**, where the edges of the basal plane develop into ridges during sublimation, which delimit the "scallop" morphology of the prims faces (Nair et al., 2018; Magee et al., 2014; Chen et al., 2017). Hence, by following the sublimation process, the smooth ice surface, which shows few contrast variations, transforms into a rugged morphology with bright high-contrast features. These contrast changes are intrinsic (Nair et al., 2018), i.e., not related to the detector performance, which would result in a change in brightness over the whole image.

Closer inspection, but carefully avoiding beam damage (see Supplement), reveals that the bright ridges are entirely composed of sub-μm features, aligned along edges or grain boundaries. Baker et al. found similar features at much lower temperature (Baker et al., 2003), however, they show a relation between the appearance of these features and the presence of contamination (in natural ice samples). This impurity effect has been studied by several authors who suggest that ions (from salts or acids) or dust particles in the crystals may be the underlying cause for the macroscopic structures that develop during sublimation (Cross, 1969; Cross, 1971; Mulvaney et al., 1988; Cullen and Baker, 2001; Rosenthal et al., 2007). In contrast, we exclude contamination; the few particles we found would by far no suffice to align at all ridges during sublimation (see Supplement). Our bright ridges stem from localized high emission currents, and must be linked to the above-mentioned asperities that appear during evaporation. We explain their presence not by contamination as in natural samples (Baker et al., 2003), but by the coexistence concave "scallop"-like surfaces (Libbrecht, 2005) with convex asperities (Chen et al., 2017). The concave surfaces form by sublimation of faces of droxtal-like crystals (Libbrecht, 2005), the convex ones are the tip-like asperities at the ridges, which delimit the concave areas (their curvature radii can reach the nanoscale, see Supplement). Whenever we zoomed into details of the asperities, we observed artefacts from beam damage (e.g. melting); it is thus likely that the smallest features are





in the range of the beam diameter (several nm). The reason should be the intrinsic non-equilibrium scenario during sublimation (see also the example of GaN during non-equilibrium growth (Sun et al., 2011)), which allows the coexistence of positive and

negative curvatures. This is based on balancing energetics with surface area, which explains the presence of a small fraction of highly curved surfaces (of an overall small area fraction) (Sun et al., 2011).

We found similar strings of bright dots on what is left of the facets of isolated crystals shown in **Figure 5E** (before) and **5F** (after sublimation). As already discussed above, these facets had - during the merging of crystals - grown very fast and

displayed a high density of grain boundaries. Such structures are often hidden below the apparently smooth surface of polycrystalline ice films, on which we will focus below.

### 3.5 Sublimation of ice films

The principal features of the sublimation process of polycrystalline ice are very similar to the case of isolated crystals discussed

above, as long as the parameters are not changed to much lower temperatures (Pfalzgraff, 2010; Magee et al., 2014). However, the different morphology of the films, i.e. rather flat ice crystallites, separated by grain boundaries (dark lines) (**Figure 6A**) first result in "etching", a widening of the grain boundaries. After 47 s (**Figure 6B**), the ice film is already slightly etched due to sublimation. Following the theory by Barnes et al. (Barnes et al., 2003), the etching of grain boundaries by sublimation should create ever widening channels. The fact that the etched grain boundary channels are much wider than the boundary

itself, an effect used to easily visualize the boundaries by optical microscopy (Barnes et al., 2003). We followed the etching in real time at -10°C, where the widening is in the range of mm/min, similar to findings by Chen et al. (Chen et al., 2017). The channels simply move vertically, together with the evaporating ice surface. When the grain boundary plane is not exactly normal to the surface, the channels no longer mark the initial boundary location, and also move laterally. As the sublimation proceeds further (from **Figures 6B** to **6C**) features reminiscent of droxtals or isolated crystals appear (Nair et al., 2018; Cross,

1971; Chen et al., 2017). Quantification of this process is rather difficult. However, the rates of sublimation are of the order of hundreds of μm/min, which appears to compare well with several images shown by other authors under similar conditions of temperature and pressure (Nair et al., 2018; Chen et al., 2017).

**Figure 6C** also shows that at the later stages of sublimation (t = 124 s) the grain boundary contrast appears inverted, i.e. white,

as for sublimation of an isolated crystal (see **Figure 5**). In the discussion of Figure 5, this phenomenon is synchronized with the appearance of sharp features. A more detailed view of this phenomenon is described below. In addition, as sublimation proceed (see **Figures 6B to 6C**) the ice asperity density increases. In polycrystalline films, asperities form during sublimation at locations where grains were located (Thompson, 2000). Hence, this increase in asperities indicate an increase in ice grain density during the film formation. This is a common feature of a polycrystalline film for which the grain structure resulting

from the nucleation, growth, and coalescence processes during the early stage of growth is retained at the near surface of the





silicon wafer; and where through competitive growth, grains with preferred orientations are favored. The result of the competitive growth is seen in the film morphology reported in Figure 6A, where fewer larger grains are visible at the surface of the film.

The change of contrast in the ESEM image during sublimation of polycrystalline and single crystals can be explained by taking a closer look at the ice morphology during sublimation. **Figure 7A** to **7D** shows how tens of µm large ice features change during sublimation. The dark features corresponding to the grains are becoming concave during sublimation. These concave features are especially well imaged next to bare surface areas, similar to our findings for sublimation of single crystals. In the final stages of sublimation, where the substrate exerts some influence on the ice morphology, the features meet in sharp ridges
and tips/asperities, with radii down to the nanometer range (similar to the features at evaporating brine (Yang et al., 2017)). The interaction with the solid substrate has some stabilizing effect on these features, which could be mechanical (immobilization), but also thermal (good heat conduction against beam damage). However, the sharp tips exist also on the ice surface, at -20 °C and 100 Pa (**Figure 7 E and F**). The appearance of inverted contrast in **Figure 6C** indicates that these sharp features also develop without apparent interaction with the solid substrate.


A specific problem here, but also more general, is contamination. We exclude contamination, and possible pinning of ice asperities at contamination spots, by testing growth and sublimation on large scales (see Supplement). **Figure 7C** and **7D** show some examples, which are reminiscent of Cross' images for apparently thicker films (Cross, 1969; Cross, 1971). However, we have repeatedly found this morphology at up to -10°C, at much higher temperature than in Cross' work (his equilibrium
pressure <0.01 Pa translates to <-90 °C). The smallest radii might be well below 100 nm, but beam heating effects (see Supplement) hinder a closer observation. Whatever the nanoscale structure might be, the very appearance of the needles/spikes requires that the curvature on the surface changes from predominantly convex on the grains (except the relatively small area of the grain boundaries) to a large fraction of concave features, as also observed in various ice crystal habits (Libbrecht, 2005).

**4. Conclusions**

We investigated the growth and sublimation of isolated ice crystals (hexagonal plates or columns, and prisms), grown in random directions on oxidized Si wafer, and of polycrystalline films, by real-time, *in situ* ESEM. We found that crystalline substrates are not required to grow isolated crystals, except if epitaxial orientation is sought. We presented an analysis of the observed microscale surface morphology, including hexagonal prisms and droxtals (droplet crystals), and of polycrystalline
ice films. The observed growth scenarios are generally compatible with reports based on SEM studies (mainly at lower temperatures), with recent ESEM results, and with optical microscopy, which is usually carried out at ambient pressure.



Our spatial and temporal resolution revealed new details on dynamic phenomena during ice growth. Specifically, we demonstrate how merging of isolated crystals, the development of uneven surfaces and of grain boundaries, ultimately creates

films with typical polycrystalline ice surfaces. We thus documented the transition from single crystal to polycrystalline morphologies for the full scenario of vapor growth, i.e., from the initial nucleation over coalescence of crystallites, to the final smooth surface, which is dominated by grain boundaries.

The appearance of small "pores" in grain boundaries has to our knowledge not been documented before and are interpreted as

being truncated multiple grain boundaries, which appear at the interface of two merging crystallites having multiple imperfections at their surfaces. Their coalescence results in the formation of a grain boundary featuring additional defects in the form of a string of dark "pores", each of only 1-3 μm diameter.

The "etching" procedure is an important technique in optical ice microscopy, and relies on imaging grain boundaries during

sublimation. We characterized the evolution of the ice morphology at the micro- and nanoscale, by following the evolution of rather complex micropatterns towards a concave "scallop" morphology. Each structure is lined by sharp ridges, which are composed of nanoscale asperities. Through careful control experiments, we know that contamination plays a minor role in their formation, different from natural samples that contain salts.

Further experiments call for technical improvements to prevent beam damage especially since the nanostructures are prone to heating and thus melting. We suggest WetSTEM techniques, more sensitive detectors, and improved electron sources. A different issue is moving towards more realistic conditions for atmospheric and geoscience research. ESEM setups allow only for pure water vapor, but no additional gases. This limits investigation of environmentally relevant conditions, including air, and impurities. Furthermore, in practice, only low supersaturation is accessible, which means that solid/liquid interfaces are

only accessible at rather low pressure, such that for example typical snow formation cannot be investigated. Local dosing of water vapor could be a simple alternative to achieve the required very high supersaturation, at least when highly dynamic growth scenarios are desired.

Further development of ESEM would afford more flexibility in experimental design and could include lower temperatures

(down to –40 ℃) and improved pressure control. A challenge is accessing the water/ice phase transition at relatively high pressure values, and generally reaching the extreme values of temperature and pressure, and fast cooling rates. Ambient pressures would require a pressure cell with much loss of resolution. Another possibility would be the addition of traces of ice nucleators such as minerals (Vetráková et al., 2019) or biogenic matter nucleation (Cascajo-Castresana et al., 2020), to research cloud nucleation. An in-situ method such as the one proposed in our study would allow to observe changes in morphology,

determine the role of grain boundaries as well as impurities in ice in real time. Moving towards mixtures, the interaction of



carbon dioxide/water vapor with dry ice/water ice, relevant for the Martian atmosphere, can be in the range of an advanced instrument.

**Author contributions**

All authors designed the experiments, MCC and AMB carried them out, all authors evaluated the results and prepared the manuscript.

**Competing interests**

The authors declare no competing interests.

**Data availability**

Data repository York University under doi.org/10.5683/SP2/S2HBAR

**Acknowledgments**

We are indebted to C. Tollan, CIC nanogune, for technical help with the ESEM. We are grateful to Prof. H. Mader, Bristol University for discussions on ice veins. We thank Prof. A. Chuvilin, CIC nanoGUNE, for discussing beam damage and contrast mechanisms. We acknowledge funding from FEI (Eindhoven, NL), Elkartek 2015 and 2019 (Basque Government), the Spanish
MINECO grants PID2019-104650GB and MAT2013-46006-R, Basque Government Grant Proyecto de Investigación PI2013-57, and from the Maria de Maeztu "Units of Excellence" Programme MDM-2016-0618 (MINECO). S.M. acknowledges York University and CIC nanoGUNE for their support during her sabbatical and subsequent visits.

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

**Figure captions**

**Figure 1. Ice growth examples recorded at 20 kV. (A) Isolated crystals grown on an oxidized silicon wafer substrate. Note that some grains have started to touch producing polycrystalline ice (grain boundaries are visible). ESEM imaging conditions: $P$ = 135.1 Pa, $T$ = -18.4ºC, $P_{sat}$ = 120.4 Pa, $h$ = 1.12. (B) Individual hexagonal crystal of ice on a clean silicon wafer. The basal plane is nearly parallel to the substrate. The outer rim is bright, an effect that might indicate charging. ESEM imaging conditions: $P$ = 127.1 Pa, $T$ = -18.7ºC, $P_{sat}$ = 117.0 Pa, $h$ = 1.09.**

**Figure 2. Ice crystals on an oxidized silicon wafer substrate, imaged at 5 kV during growth. Droxtal-like (1-4) and pillar (5) morphologies are clearly identifiable. ESEM imaging conditions: $P$ = 148.1 Pa, $T$ = -18.2ºC, $P_{sat}$ = 122.7 Pa, $h$ = 1.21.**

**Figure 3. Isolated crystals merging during growth. As soon as the crystals touch, the planes in contact show polycrystalline topography with a large density of grain boundaries. Two growth spirals are traced by white dashed lines. ESEM imaging conditions: (A), (B), (C) $P$ = 130.2 Pa, $T$ = -17.9°C, $P_{sat}$ = 126.2 Pa, $h$ = 1.03; (D) $P$ = 126.8 Pa, $T$ = -18.3°C, $P_{sat}$ = 121.5 Pa, $h$ = 1.04. The images show the same area of the sample.**



**Figure 4. (A) Hexagonal pillars of isolated crystals and polycrystalline ice films grown on an oxidized copper surface. ESEM imaging**

**conditions: $P$ = 135.1 Pa, $T$ = -17.8°C, $P_{sat}$ = 127.4 Pa, $h$ = 1.06. (B) Typical polycrystalline ice surface after growth and coalescence**

**of the crystals on Si. The grains appear bright with typical shading effects. They are limited by 1 to 2 µm wide dark bands, the grain**

**boundaries. Three boundaries meet at angles of mainly 90º to 120º. Each grain boundary consists of aligned pores of ≈ 1-3 µm in**

**diameter. $P$ = 438.4 Pa, $T$ = -4.8ºC, $P_{sat}$ = 409.0 Pa, $h$ = 1.07. (C) Polycrystalline ice surface during growth. Circular pores aligned**

**along grain boundaries. $P$ = 287.3 Pa, $T$ = -10.3ºC, $P_{sat}$ = 253.3 Pa, $h$ = 1.13.**

**Figure 5. Sublimation sequence of hexagonal crystals. (A) to (D) shows the basal plane of a single crystal. ESEM conditions: (A) $P$ =**

**180 Pa, $T$ = -15.2ºC, $P_{sat}$ = 162.4 Pa, $h$ = 1.11; (B) $P$ = 139 Pa, $T$ = -18.2ºC, $P_{sat}$ = 122.7 Pa, $h$ = 1.13; (C) $P$ = 123 Pa, $T$ = -19.1ºC, $P_{sat}$ =**

**112.6 Pa, $h$ = 1.09; and (D) $P$ = 97 Pa, $T$ = -20.1ºC, $P_{sat}$ = 102.4 Pa, $h$ = 0.95. All images have the same scale (see C). The time elapsed**

**since the onset of sublimation is given in each figure. (E) Merged crystals, -18.3ºC, $P$ = 126.8 Pa, $P_{sat}$ = 121.5 Pa, $h$ = 1.04 (F) The**

**same area during the last stage before complete sublimation (6 min later). The bright grey background is the pure silicon wafer**

**surface; all other features are ice crystals. Some facets are decorated with bright lines (strings of bright dots). ESEM imaging**

**conditions: $P$ = 130.6 Pa, $T$ = -17.1ºC, $P_{sat}$ = 136.1 Pa, $h$ = 0.96.**

**Figure 6. A thick polycrystalline ice film during sublimation. The grain boundaries widen substantially. The last image appears to**

**show contrast inversion of the grain boundaries. ESEM imaging conditions: (A) $P$ = 253.4 Pa, $T$ = -10.2ºC, $P_{sat}$ = 255.5 Pa, $h$ = 0.99;**

**(B) $P$ = 230.9 Pa, $T$ = -10.1ºC, $P_{sat}$ = 257.8 Pa, $h$ = 0.90; and (C) $P$ = 232.8 Pa, $T$ = -10.1ºC, $P_{sat}$ = 257.8 Pa, $h$ = 0.90. The time from the**

**onset of sublimation is given in the figures. All images have the same scale (see A).**

**Figure 7. A thick polycrystalline ice film during sublimation. Droxtal-like shapes appear, reminiscent of isolated crystals. In the final**

**stage, needles/spikes develop. The smallest radii of such structures is below 10 nm. ESEM imaging conditions: (A) $P$ = 137.0 Pa, $T$**

**= -15.0°C, $P_{sat}$ = 165.4 Pa, $h$ = 0.83; (B) $P$ = 147.3 Pa, $T$ = -13.5°C, $P_{sat}$ = 189.8 Pa, $h$ = 0.78; (C) $P$ = 142.0 Pa, $T$ = -12.1°C, $P_{sat}$ = 215.6**

**Pa, $h$ = 0.66; and (D) $P$ = 129.0 Pa, $T$ = -9.7 °C, $P_{sat}$ = 267.1 Pa, $h$ = 0.48. The time from the onset of sublimation is given in the figures.**

**All images have the same scale (see C). (E): -20°C. The ice on the right hand side part is much thicker than on the left, and shows**

**the typical concave "scallop" shape on the 10 µm scale. This is replicated also in the sub-µm scale. The ridges (white lines) and tips**

**appear very bright due to the high local emission current. (F) Zoom into (E).**




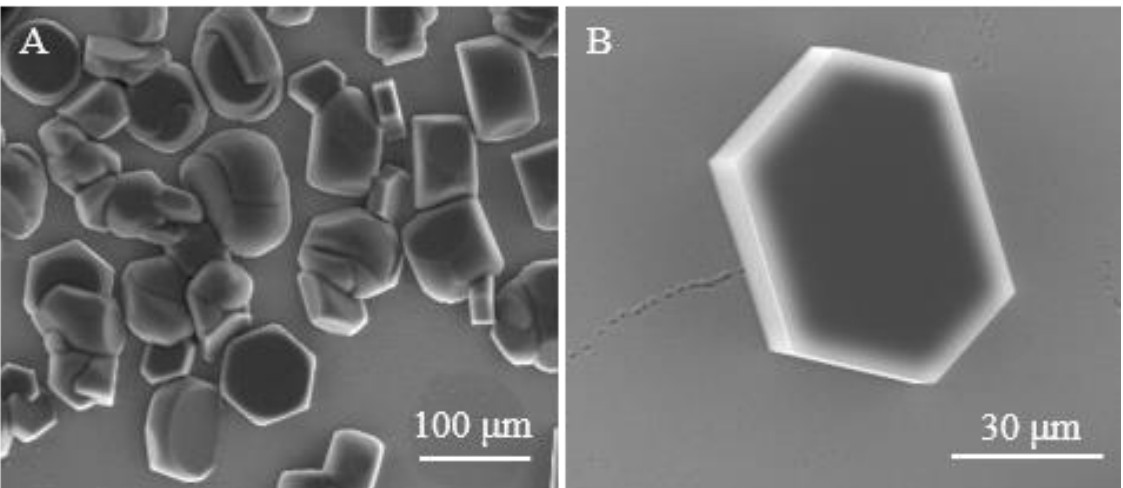

**Figure 1**




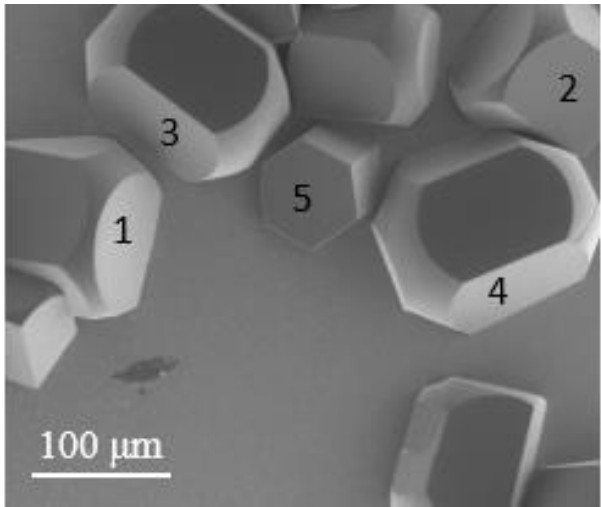

**Figure 2**




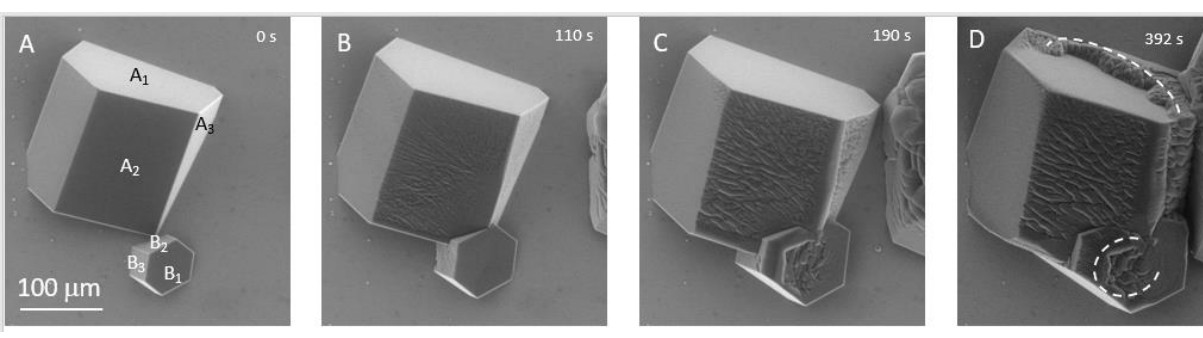


**Figure 3**





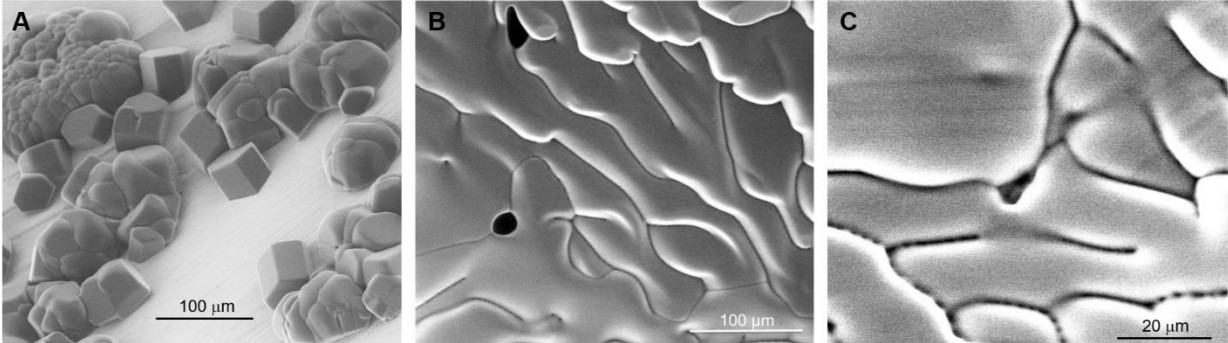


**Figure 4**


**Figure 5**




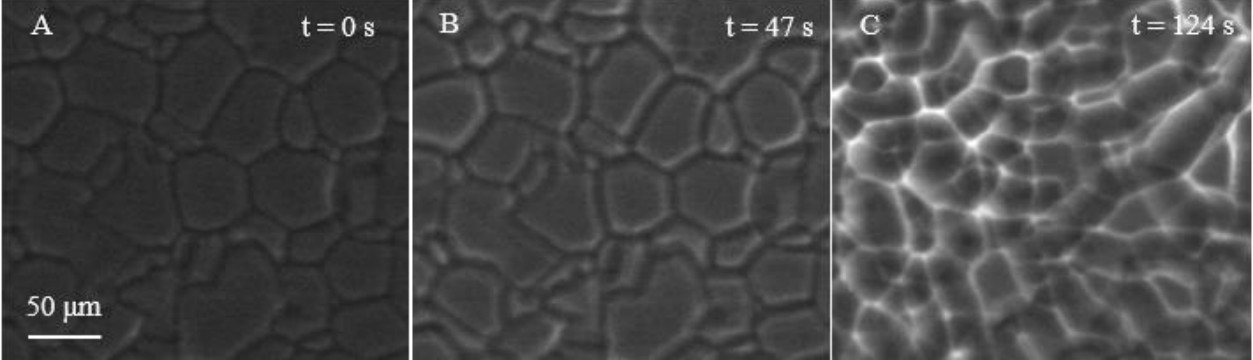

**Figure 6**



**Figure 7**






**TOC (table of contents) figure**