# Peer review of "The ice-vapor interface during growth and sublimation"

_Atmospheric Chemistry and Physics, 2021_

## Author Comment (AC1)

https://doi.org/10.5194/acp-2021-335-RC1

Reply to referee 1

This was our first attempt at studying the ice-vapor interface using ESEM and we appreciate the time referee 1 took to look at the paper in detail and to formulate helpful comments. Below are our replies to these comments. We kept these replies very short because the modifications to the manuscript are extensive.

*1) Throughout the manuscript: Description of temperatures*

*In this manuscript, just beside abstract, all values of temperatures are written "without minus signs": e.g. -10 °C --> 10 °C. I (a crystal growth physicist) am not familiar with manners in the field of atmospheric science. However, for me, such description of temperatures looks very strange. If this is not a manner in atmospheric science, the authors need to describe in a correct way.*

This was a software problem, and the omission was corrected throughout the text.

*2) Lines 33-39 and throughout the manuscript: Droxtals*

*As the authors explained, "droxtals" are formed by transformation of "supercooled water" into spherical ice crystals (i.e., by melt growth). The rounded morphology of the droxtals is determined mainly by a rounded outer shape of a liquid water droplet. In contrast, all ice crystals in this paper were formed by transformation of "supersaturated water vapor" (i.e., by vapor growth). For the details of mechanisms that may determine the outer shape of ice crystals formed by vapor growth, see comment 10). The formation mechanisms of rounded ice crystals by vapor growth and those by melt growth are fully different. Therefore, although the ice crystals with rounded corners and rounded ridges show the shape similar to that of droxtals, the authors should not call them "droxtals" (they are not analogous to droxtals), in order not to give misunderstanding to readers.*

Thank you for pointing this out. Although the resulting morphology is similar, we were not aware of the fact that the term "droxtal" is restricted to growth from liquid. We remove our reference to droxtals to describe our rounded edge crystals grown from water vapor. We refer to droxtals now exclusively to highlight differences, and avoid confusion with growth from supercooled water (Lines 35 and 36).

*3) Lines 59-65: Heteroepitaxial substrates*

*I believe this paragraph is not necessary, because the authors only used substrates whose outer surfaces were amorphous SiO2, and also because this manuscript is not a review paper.*

We removed the paragraph discussing heteroepitaxy.

*4) Lines 67-86: Changes in morphologies with temperature*

*These paragraphs look very redundant. If the authors show a wide variety of morphologies of ice crystals in this study, such detailed explanations of habit changes are necessary. Instead of such redundant explanations of the habit changes, I believe that the authors need to prepare introduction that are closely related to the authors' new findings in this study: e.g. the growth of rounded ice crystals in vapor, the formation of pores in grain boundaries, and so on.*

These paragraphs now Lines 52 to 57 have been shortened extensively. We have indeed refocused our discussion on our new findings (see discussion for reference).

*5) Lines 97-102: Objectives and what's new*

*In somewhere (probably at the end) of the introduction section, the authors need to clearly explain what is new in this study, otherwise the present introduction is too general and vague. In addition, in the present introduction (or in somewhere of this study), there is no explanation how the authors' ESEM studies are different from those performed by other researchers (in particular, from the technical viewpoint). Please include them.*

See Lines 76 to 81 for objectives and comment on the novelty of the work. We are the first to report all setup details, including the problematic control issues of temperature and pressure, beam damage on the nanoscale and on the microscale, and to assess potential contamination issues. The relevant data compiled in the Supplement and in our repository (doi.org/10.5683/SP2/S2HBAR) would be fit to assemble a technical paper on its own.

*6) Lines 115-122: The control of water vapor pressure in ESEM*

*The authors should explain "the control and measurement" of water vapor pressure more in detail. Here, the authors also need to explain "experimental errors" in determining water vapor pressure P and supersaturation h. In results and discussion section, the authors showed precise values of water vapor pressure: e.g. P=130.6 Pa. But for me it looks difficult to determine.*

Section 2.1 was edited extensively see Lines 88-99 and Lines 118 to 119 (see also Supplement and new Figures S9 and S10)

*7) Lines 169-180: The bright contrast*

*The authors should explain the effects of negative charges on the growth of ice crystals. If such effects are unclear at present, the authors need to clearly explain so.*

See Lines 168 to 171. In order to investigate the effect of charging on ice growth (and sublimation), these processes were studied with the beam focused (and scanning) and not focused on the areas of interest. We observed that the presence of the beam resulted in no morphological changes in the ice crystal or film features. However, prolonged scanning with a high beam voltage induces beam damage and carbon deposition, which may alter the growth and sublimation processes (see Supplement).

*8) Lines 182-189: Nucleation of ice crystals on substrates*

*I do not believe this paragraph provides useful information. The authors should delete this paragraph.*

This paragraph was deleted.

*9) Lines 191-198:*

*(1) The authors wrote "When the growth is monitored over time a growth rate of ~100 nm/s is observed". I suppose this is the "normal growth rate" of prism faces. Please explain properly.*

The growth rate can reliably only be determined for changes parallel to the surface. Thus measured values are reported in the Supplement, inside Fig. S2D.

*(2) The authors also wrote "Our conditions are best". I cannot understand that the authors' conditions are best from what kind of viewpoint.*

This was a language issue and it was corrected. See line 178.

*(3) The authors need to explain why the growth rate vary substantially over time. Why did the growth rate increase with time?*

Although we found this in several measurements, we lack sufficient statistics. Moreover, other reports are unspecific, or not sufficiently detailed. The literature on growth at much lower temperatures gives precise values, but is not relevant for our parameters. So we decided to remove this statement.

*10) Lines 213-221: Roughening*

*Here, the authors are fully misunderstanding the roughening mechanisms of faceted ice crystals in vapor. There are three different mechanisms that were revealed so far.*

*(1) Roughening during sublimation: Nelson (1998) reported that faceted ice crystals are rounded "during sublimation". During growth, crystal faces with slower growth rates are developed with time. Whereas, during sublimation, crystal faces with faster sublimation rates are developed with time: in other words, crystal faces with slower sublimation rates disappears. Therefore, Nelson said that sublimation rates of basal and prism faces are slower than high-index (rough) faces. The authors' study is on ice crystals "only during growth". Therefore, this is not the case.*

*(2) Thermal roughening at a temperature very close to the melting temperature: Elbaum (1991) revealed that prism faces of ice crystals in air are rounded (disappeared) by "thermal roughening" at temperatures higher than -2 °C. In this case, temperature is extremely high. Therefore, the increase in entropy (roughness) decreases the Gibbs free energy of the system, according to the famous relation deltaG = deltaH -T deltaS. Thermal roughening proceeds under supersaturated, equilibrium, and also undersaturated conditions (irrespective of water vapor prerssure). As a precursor phenomenon, the roughening of ridges can occur at temperature slightly lower than -2 °C. However, the authors' study is at -20~-10 °C. Therefore, it is difficult for me to imagine that this is the case. If the authors' low vacuum condition effectively decreases the thermal-roughening temperature from -2 to -10 °C, this paper will become a great job (but the authors need to prove it)!*

*3) Kinetic roughening under highly supersaturated conditions: The last case proceeds when supersaturation is extremely high. With increasing supersaturation, the size of a critical two-dimensional (2D) nucleus decreases significantly. When the size of a critical 2D nucleus becomes equal to or smaller than a size of a water molecule, kinetic roughening occurs. In other words, even one water molecule can become a 2D nucleus. Therefore, crystal surfaces become very rough. However, the authors' growth experiments were performed under relatively small supersaturation. Therefore, probably this is not the case. If the authors' low vacuum condition significantly decreases the critical supersaturation necessary for kinetic roughening, this paper will become a great job (but the authors need to prove it)!*

*In conclusion, at present, I cannot identify the reason why ice crystals growing under low supersaturation become rounded. The elucidation of this mechanism will become an important separate study. In addition, as explained in the comment 2), droxtals are formed during "melt growth" (not during vapor growth). Therefore, the droxtals cannot be analogous. The authors need to fully rewrite the relevant discussion.*

We have re-analyzed some of our data and concluded that roughening is indeed occurring only during sublimation. Whenever round features are observed during growth, the objects were not isolated crystals and/or had been subjected to growth and partial sublimation. We are providing evidence for this both in the text (sections 3.4 and 3.5) and the Supplement (including two new extensive time series/movies, Figure S9 and S10, plus text. Hence, our results are in line with other observations and with the theory. We are very much indebted to the referee to provide such detailed criticism.

*11) Lines 222-229: Changes in shapes with time*

*The authors wrote that crystals 1 and 2 have changed their shape significantly during growth. If this is true, I strongly recommend that the authors should show the time-course of ESEM pictures which demonstrate how the shape of the crystals was changed as time elapsed, because such time-course will give a strong clue for elucidating the mechanism of the roughening, as explained in the comment 10).*

The original text was revised, please see lines 205-209.

*Here also, "droxals" formed by melt growth cannot be analogous to the authors' crystals formed by vapor growth.*

This was corrected, see comment 2).

*12) Lines 238-245: Growth rates of crystal faces*

*In this paragraph, I could not understand what the authors wanted to explain. Did the authors mean that different crystal faces (even with the same crystallographic indices) exhibited significantly different growth rates depending on spatial configurations? If yes, the authors should concretely show the relation between the growth rates and their spatial configurations by adding the values of the growth rates in the ESEM picture. Otherwise, this paragraph does not give any meaningful information. In addition, the droxtal cannot be analogous.*

*The authors wrote "some faces (.....) expand only very slowly at their edges, and practically not in their normal. Here also, I could not catch the meanings. What does "not in their normal" means?*

We measured local growth velocities at isolated and merging crystals shown in Figure 3. Measured values are reported together with Figure S1 (isolated crystals) and S2D (merging crystals) in the Supplement. For our set of parameters (moderate supersaturation) we obtain values up to 200 nm/s. The values can be separated into max 100 nm/s at the six edges of the basal plane, and 50 to 150 nm/s for some prism faces. Some edges, though, appear to be at least partially pinned, which is not due to contact with the substrate. However, some faces (e.g., the basal plane A1 in Figure 3) expand only very slowly at their edges, and practically not in their normal (90º from surface plane). This could contribute to the rounded shape, where the basal plane is almost preserved, while all edges become progressively rounded until they merge. Even then, the center of the basal plane is preserved, as seen in Figure 4A.

*13) Lines 246-257: The formation of grain boundaries during the merging of neighboring crystals*

*(1) The authors wrote that these grain boundaries are formed very fast and appear even hundreds of μm away from the area of contact between two crystals. The speed of the formation of grain boundaries are just determined by the growth rates of mother crystals during their merging, and the length of grain boundaries shows the distance of the overlap of*

*mother crystals. Therefore, I believe that the speed and distance of the grain-boundary formation have no meaningful information.*

*(2) I could not understand the contents written on the lines 250-253. What does the sentence "The interface between planes with different orientations then becomes the nucleation site for additional molecular layers that are not matching with the structure found initially" mean? Although I am an expert of crystal growth physics and ice crystals, I have never known such phenomenon. On an interface (grain boundary) between adjacent crystals with different orientations, large amount of strain energy is formed, and then many dislocations (avalanche of dislocations) are formed to decrease the strain energy. In an extreme case, many dislocations thus formed might be able to promote the formation of molecular layers whose orientations are different from that of a mother crystal. But as far as I know, no study has so far yet proved the presence of such phenomenon experimentally. If the authors further want to claim the presence of "the molecular layers with a different structure", the authors need to cite other studies and also need to show experimental evidence, such as ESEM pictures, diffraction patterns, etc. I recommend that the authors should remove this claim.*

*With respect to the corrugations shown on the face A2 in Figures 3B-3D, the authors can write something as follows: "After the contact of adjacent crystals, molecular layers that were newly formed on a crystal surface may contain many dislocations. The corrugations shown on the face A2 in Figs. 3B-3D may show the effects of such dislocations included in the newly-grown molecular layers."*

Lines 224-237:
Part(1): We corrected all relevant phrases together with distinguishing grain boundaries and grooves, and describe the "rough" surface now as "corrugated". We would like to continue avoiding the wording "rough" to avoid confusion with roughnening.

Part 2: We are in agreement with the referee's comment and text has been edited accordingly.

*14) Lines 259-265: A process analogous to the onset of spiral growth*

*I cannot accept this paragraph. If the authors claim that it was possible to record a process analogous to the onset of spiral growth, first the authors need to show ESEM pictures of "a much clear spiral pattern" and "its time course", from which readers can understand how the spiral pattern was developed as time elapsed. Otherwise, I recommend that the authors should fully delete this paragraph because of the following reasons.*

*The authors cited the studies reported by Thürmer's group (2013) and Sazaki's group (2010, 2014, 2014). These studies observed the spiral growth of "elementary steps" of 0.37 nm in thickness (corresponding to the size of one water molecule) by SPM and advanced optical microscopy. A Burgers' vector of a screw dislocation on a basal face shows a size of integral multiple of 0.74 nm (a unit-cell size in the crystallographic c-direction). Strain energy formed by a screw dislocation is proportional to the square of the size of the Burgers' vector. Therefore, with increasing size of the Burgers' vector, the strain energy formed by the screw dislocation increases drastically, and hence such giant Burgers' vector becomes implausible. This is the reason why the presence of spiral steps that are visible by ESEM is implausible.*

*There can exist another case in which original spiral steps were of elementary height, but they were bunched during the growth. Then bunched spiral steps could be observed by ESEM. In this case, the authors need to show the time course of much clearer ESEM pictures, as explained above.*

*The conclusion that I wanted to tell the authors here is the absence of a process analogous to spiral growth.*

Thank you for pointing this to us. We have corrected the text accordingly (Lines 232 -237)

*15) Line 284: The width of a grain boundary*

*The width of a grain boundary is at the molecular scale (smaller than nm). The width that was 2 μm on average is the "apparent width" of a "groove (opening)" of a grain boundary. In the case of the merging of the corrugated face (the face A2 in Figures 3B-3D) and a flat face, the width of a grain boundary is the distance between the flat face and the top of the convex curved-surface of the corrugation. The distance between the flat face and the bottom of the concave curved-surface of the corrugation is just a diameter of a pore that was not closed during the growth process.*

References to "grain boundaries" that in fact referred to microscopic grooves (dark bands) were changed throughout the text, with the exception of very few instances, where we intended to address the atomic-scale interface between two grains.

*16) Lines 285-287: Relative angles of grain boundaries and veins*

*The differences between the veins and the grain boundaries that the authors observed are not only the presence/absence of brine but also the presence/absence of the "grain growth" driven by grain boundary energies. The authors observed the grain boundaries just after the formation of the grain boundaries by merging adjacent crystals. In contrast, ice grains that include the veins experienced a certain amount of time period, during which the grain growth proceeded. This is the reason why the veins show more-stable relative angles of 120°.*

We adjusted text accordingly by mentioning the slow equilibration process of the veins (see Lines 259-262).

*17) Lines 293-294: The width of grain boundaries*

Please See comment (15).

*18) Lines 310-314: Temperature*

*The authors wrote "given the low temperature we applied during growth, coarsening towards a smooth shape should not readily occur". This is fully wrong: the coarsening should occur, and the circular shape of the pores is common. High/low temperature is determined by the difference between the experimental temperature and the melting temperature (0 °C). The authors' experimental temperature -20~-10 °C reaches 93~96% of the melting temperature. Therefore, they are very high temperature close to 0 °C. Under such high temperature, water molecules in an ice crystal can be relatively-freely moved and rearranged.*

We agree with the comment and remove reference to coarsening.

The "low temperature" is correct w.r.t ESEM experiments, but to avoid confusion for readers who work at temperatures well below -40ºC, where neither supercooled water nor the quasi-liquid layer should play a role, we decided to avoid this term.

*19) Figure caption of Fig. 5 and relevant main text*

*In the caption, supersaturations in the panels A-D are h=1.11, 1.13, 1.09, and 0.95, respectively. According to this caption, the ice crystals in A-C were in supersaturated water vapor (growth occurred), and only the ice crystal in D was in undersaturated water vapor (sublimation occurred). According to the main text, I suppose that the crystals shown in A-C were also in undersantruated water vapor, and they were sublimated, because the values of t in the ESEM picture show the elapsed time after the sublimation started. The authors need to revise appropriately.*

See reply (6) and also section 2.1. We added text to clarify this issue, i.e. " This automatic pressure control balances water vapor supply and removal by pumping, and allows to reach h > 1 only for short times (at best some minutes). Due to the size of the specimen chamber, most of the vapor is at ≈+20 ºC, but due to the very small flow rate, it attains the substrate temperature when it approaches the Peltier stage/sample assembly."

*20) Lines 329-330:*

*The authors wrote "An unusual observation is reported in Figure 5B, where the edges of the basal plane develop into ridges during sublimation". I cannot understand this description at all. The authors need to revise appropriately.*

The description is now improved: "In Figures 5B and 5C, ridges and protrusions are clearly visible and often appearing as bright high-contrast features (white) (see Gonda and Sei 1987 for analogous behaviour observed using optical microscopy). This process extends to the prism faces and becomes more pronounced in Figure 5C. As the sublimation proceeds, the ridges delimit concave features seen mainly on what is left of the prism faces. These contrast changes are intrinsic (Nair et al., 2018), i.e., not related to the detector performance, which would result in a change in brightness over the whole image."

*21) Lines 348-352: The intrinsic non-equilibrium scenario during sublimation*

*Readers (including me) cannot understand the intrinsic non-equilibrium scenario without reading several references cited by the authors. Therefore, the authors need to briefly explain the scenario in the main text.*

The non-equilibrium condition (i.e. growth or sublimation) was simply used to further stress the surprising observation of convex and concave shapes. We corrected this by citing Gonda and Sei 1987 who report growth ad sublimation of hollow ice crystals, which makes the same point without explicitly invoking non-equilibrium (Lines 314 - 319).

*22) Lines 353-356: High density of grain boundaries*

*Here, the authors are discussing the origin of the strings of bright dots that appeared during the sublimation of isolated ice crystals. However, the authors explain that these facets had grown very fast and displayed a high density of grain boundaries. The term "grain boundary" means a boundary between adjacent crystalline grains with different crystallographic orientations. In other words, there is no grain boundary inside a single crystal. Therefore, the authors' description is wrong. The authors should replace "grain boundary" with other terms, such as aggregates of dislocations (or subboundary), that can be included in a single crystal.*

Grain boundaries were referred to by mistake and this was corrected (see Lines 301 - 304).

*23) Line 379: Ice grain density during the film formation*

*Here, I suppose that at the very beginning of the growth of the polycrystalline ice thin film, the number density of ice grains (crystals) was high. However, as the thin film grew, the number density of ice grains became smaller because of the grain growth. Then, the number density of ice grains in the vicinity of the substrate is higher than that on the surface of the polycrystalline thin film. If my supposition is correct, the authors should replace "an increase in ice grain density during the film formation" with "a decrease in ice grain density during the film formation" or "an increase in ice grain density during the film sublimation".*

Corrected (See Lines 337-338)

*24) Line 420: Truncated multiple grain boundaries*

*I cannot understand what "truncated multiple grain boundaries" means. The authors need to revise properly.*

We removed "truncated multiple grain boundaries" and described the effect as an interfacial process, occurring between two merging crystallites having multiple imperfections (see Lines 380-383 and drawing supp. info Figure S4C).

---

## Author Comment (AC2)

https://doi.org/10.5194/acp-2021-335-RC2

Reply to referee 2

*The authors observed the growth and sublimation of ice crystals using ESEM at temperatures between -10 and -20°C (n.b. the preprint has many missing negative signs) and pressures with over-and undersaturation. They describe their observations by ESEM images.*

This was a software problem, and the omission was corrected throughout the text.

*Pressures below the typical surface pressure occur in the atmosphere, and the paper may be therefore relevant for the formation of ice crystals in clouds. The paper investigates with similar methods but at lower temperatures and different pressures ice growth and sublimation as in Chen et al. (2017).*

*The paper is of a very descriptive nature, comparing the observations with literature. The interpretation is based on comparing specific observations with other authors. The observations are described in detail. However, quantitative data (e.g. detailed figures on ice growth and sublimation rates of repeated experiments) are missing.*

We agree that such data would be very valuable.

We measured local growth velocities at isolated and merging crystals shown in Figure 3. Measured values are reported together with Figure S1 (isolated crystals) and S2D (merging crystals) in the Supplement. For our set of parameters (moderate supersaturation) we obtain values up to 200 nm/s. The values can be separated into max 100 nm/s at the six edges of the basal plane, and 50 to 150 nm/s for some prism faces. Some edges, though, appear to be at least partially pinned, which is not due to contact with the substrate. However, some faces (e.g., the basal plane A1 in Figure 3) expand only very slowly at their edges, and practically not in their normal (90º from surface plane).

More detailed data would, however, require a completely new study that could not readily be compared to existing literature, specifically when ESEM or other imaging techniques are considered. In contrast, the literature on growth at much lower temperatures gives precise values, but is not relevant for our parameters, and it is very rarely focussed on imaging. The same arguments apply to sublimation rates.

*I could not find a substantial conclusion. The author observed a transition from single crystal to polycrystalline film as their most relevant result. Such a result may be relevant for the icing of aeroplanes. However, ice crystals in the atmosphere are usually single crystals and do not grow on a substrate.*

*The paper is clearly structured although in parts lengthy. The first paragraph of the introduction is not relevant: ice on the ground is always at quite high atmospheric pressure. The only relevance of this paper is for ice clouds. If deposition and sublimation occur with the same morphology a higher air pressure is not clear and is not discussed in this paper.*

We rewrote introduction and conclusion, which we believe are now addressing the issue better. We also mention that the atmosphere plays a decisive role for ice fields; albeit we present data at much lower pressure, our temperature and humidity conditions do apply.

*The authors seem also not aware that ice at a temperature of -20°C is actually at a very high homologous temperature (about 8% below the melting point). The "low temperature" is only on a Celsius-temperature scale.*

The "low temperature" is correct w.r.t ESEM experiments, but to avoid confusion for readers who work at temperatures well below -40ºC, where neither supercooled water nor the quasi-liquid layer should play a role, we decided to avoid this term.